# Comparison of Alere q whole blood viral load with DBS and plasma viral load in the classification of HIV virological failure

**Aabida Khan**[1¤]*, **Lucia Hans**[2], **Nei-yuan Hsiao**[1]

**1** Division of Medical Virology, Department of Pathology, University of Cape Town and National Health Laboratory Service, Cape Town, South Africa, **2** Department of Haematology and Molecular Medicine, University of Witwatersrand and National Health Laboratory Service, Johannesburg, South Africa

¤ Current address: Department of Virology Kwa-Zulu Natal, National Health Laboratory Service and University of Kwa-Zulu Natal, Durban, South Africa

* aabida.khan@nhls.ac.za

**Data Availability Statement:** All relevant data are within the manuscript and its Supporting Information files.

**Funding:** The author(s) received no specific funding for this work.

## Abstract

### Background

In remote settings, timely plasma separation and transportation to testing laboratories is an impediment to the access of HIV viral load (VL) testing. Potential solutions are whole blood testing through point of care (POC) assays or dried blood spots (DBS).

### Methods

We evaluated the performance of a prototype Alere q whole blood VL protocol and compared it against plasma (Abbott RealTime HIV-1) and DBS VL (Abbott RealTime HIV-1 DBS revised prototype protocol and Roche CAP/CTM HIV-1 v2.0 DBS free virus elution protocol). Virological failure (VF) was defined at >1000 copies/ml.

### Results

Of 299 samples, Alere q correctly classified VF in 61% versus 87% by Abbott DBS and 76% by Roche FVE. Performance varied across plasma VL categories. Alere q showed 100% sensitivity. Below 1000 copies/ml of plasma, Alere q demonstrated over-quantification, with 19% specificity. Abbott DBS had 91% sensitivity and the best overall correlation with plasma ($r^2 = 0.72$). Roche FVE had the best specificity of 99% but reduced sensitivity of 52%, especially between 1000–10,000 copies/ml of plasma. Correlation was best for all assays at >10,000 copies/ml.

### Conclusion

Variability was prominent between the assays. Each method requires optimization to facilitate the implementation of a cut-off with optimal sensitivity and specificity for VF. Although Alere q whole blood assay exhibited excellent sensitivity, the poor specificity of only 19% would lead to unnecessary switching of regimens. Thus any VF detected would need to be confirmed by a more specific assay. Both the Abbott DBS and Roche FVE protocols showed

**Competing interests:** Alere Health provided the instrument and cartridges for testing, however did not play any role in the study. There are no patents, products in development or marketed products to declare. This does not alter our adherence to PLOS ONE policies on sharing data and materials.

good specificity, however sensitivity was reduced when the plasma VL was 1000–10,000 copies/ml. This could result in delays in detecting VF and accumulation of drug resistance. Field evaluation in settings that have adopted these DBS protocols are necessary.

## 1. Introduction

In response to the HIV pandemic threat to global health, tremendous gains have been achieved towards the UNAIDS 90-90-90 goal [1, 2]. Scaling up of HIV viral load (VL) testing has been the integral part of "the third 90" goal as plasma VL remains the best predictor of viral suppression and treatment outcome [3–5]. Among the challenges facing VL access, timely plasma separation and transportation to testing laboratories remain a major barrier [6]. Sampling requires collection of a venous whole blood sample that is stable at room temperature for only up to six hours therefore requiring cold chain storage and centrifugation prior to testing [7]. Sample transport networks are often suboptimal resulting in significant delays [8].

VL testing using dried blood spot (DBS) samples bypass the issue of plasma separation and the World Health Organisation (WHO) has recommended this sampling strategy to improve the reach of VL testing services [9]. New point of care (POC) technologies such as Alere q presents a potential alternative to the current plasma or DBS VL testing in specialised VL facilities. POC testing would reduce the need for expensive laboratory infrastructure and skilled laboratory workers, and also could lower the cost of testing [10]. An easy to use, reliable and affordable POC VL would facilitate timely detection of VF, allow swift clinical decision making and reduce loss to follow up [11, 12]. The main rationale for this study was to evaluate the performance of a prototype Alere q whole blood VL assay and compare it against plasma VL and DBS VL in the classification of VF, as the need remains to evaluate POC technologies, to add to the existing DBS VL data due to the variability in testing protocols, differences in methodologies affecting sensitivity and specificity, the need for standardisation and the greater need for validated assays with regulatory approval.

## 2. Methods

### 2.1 Study design and sampling

This was a laboratory-based cross-sectional study. The primary study objective was to evaluate the performance of Alere q whole blood VL assay in the classification of VF using plasma VL as the gold standard. The secondary objective was to evaluate the performance of locally available commercial VL assays (Abbott and Roche) using DBS protocols in the classification of VF using plasma VL as the gold standard. Ethical approval was received from the Human Research Ethics Committee of the Faculty of Health Sciences at the University of Cape Town (HREC REF: 819/2014). Expedited ethics approval without the need for informed consent was received as no additional samples were collected. Strict patient confidentiality was maintained and no patient identifiers were present in data analysis. Routine plasma VL and CD4 samples from public sector antiretroviral clinics across the Western Cape Province were used in this evaluation. To ensure most of the VL analytical spectrum was covered, our convenient sampling strategy randomly selected paired CD4/VL EDTA samples where at least 50 samples of plasma VL results were from each of four categories: target not detected, <40–1000 copies/ml, 1000–10,000 copies/ml and > 10,000 copies/ml. Once a plasma VL and CD4 testing were complete, the remainder of the blood sample from CD4 testing was used for whole blood Alere q

test and DBS within 72 hours of sample receipt to prevent sample degradation. DBS was prepared by applying 75μL of whole blood to each spot on a Whatman 903 filter-paper card, dried and packed into plastic bags with desiccant sachets and stored at room temperature in boxes away from direct sunlight or heat. DBS VL were tested in batches within 1–2 months of preparation. Whole blood specimen that was used for the DBS and Alere q testing was stored at room temperature after CD4 testing.

## 2.2 Laboratory testing

The standard-of-care plasma VL testing was performed using Abbott RealTime HIV-1 assay (Abbott Laboratories, Chicago, USA) according to the manufacturer's instructions [13]. This was used as the main reference test with which samples were chosen in their respective VL categories and results from other methodologies were evaluated. The whole blood VL on the Alere q (Alere Technologies, Jena, Germany) was performed using a prototype cartridge, which detects viral RNA from 25μL of whole blood sample. The Alere q assay evaluated in this study in 2014 and 2015 was the prototype available for evaluation during this time period. Due to the low sample volume used, the limit of detection of this prototype assay is lower than the final commercially available version known as the m-PIMA™ HIV-1/2 VL test. The whole blood assay, however, has several advantages. As a POC test, capillary whole blood can be collected from a finger/heel prick directly for testing without the need for plasma separation. Operational requirements are minimal as compared to laboratory based assays as the cartridge kits can be kept at room temperature therefore no refrigeration is required. It can be used in a non-laboratory environment and can be operated on external power, with an internal rechargeable battery protecting it against power fluctuations. It does not require any user related instrument maintenance.

DBS VL protocols performed in this study were a revised Abbott RealTime HIV-1 DBS prototype protocol (Abbott DBS) and Roche CAP/CTM HIV-1 v2.0 DBS free virus elution protocol (Roche FVE) as these are the commercial platforms available in our setting. There are several workflow improvements with the revised protocol as compared to the original 'open mode' protocol which includes use of one DBS spot, number of sample tubes reduced from two to one, automated DBS eluate transfer and up to 93 DBS specimens per run therefore being more cost effective [14, 15]. The one spot Abbott RealTime HIV-1 DBS protocol was WHO prequalified and CE marked in 2016 when the limit of detection was found to be 839 copies/ml compared to the prototype protocol evaluated in this study which had a limit of detection of 1000 copies/ml [14, 16]. The performance characteristics of each of the VL assays are summarized in Table 1. DBS preparation, Alere q VL and plasma VL testing were performed at Groote Schuur Virology laboratory in Cape Town while DBS protocols were performed at HIV Molecular Laboratory at Charlotte Maxeke Johannesburg Academic Hospital.

**Table 1. HIV VL test methods.**

| Method | Assay (Abbreviation) | Sample Type | Sample Volume | Reportable Range copies/ml |
|---|---|---|---|---|
| 1 | Standard-of-care: Abbott RealTime HIV-1 [13] (Plasma VL) | Plasma | 600μL or 200μL | 40–10 million, 150–10 million |
| 2 | Point-of-care: Alere q HIV-1/2 [17, 18] (Alere q) | EDTA whole blood | 25μL | 2491–10 million |
| 3 | Abbott RealTime HIV-1 DBS revised prototype protocol [15] (Abbott DBS) | DBS | 1 x 75μL spot | 1000–10 million |
| 4 | Roche COBAS AmpliPrep/COBAS TaqMan HIV-1 Test v2.0 DBS free virus elution protocol [19] (Roche FVE) | DBS | 1 x 75μL spot | 400–10 million |

Table 1 To provides a list of the assays evaluated: sample type, volume used and reportable range. Refer to cited references for details of each assay methodology.

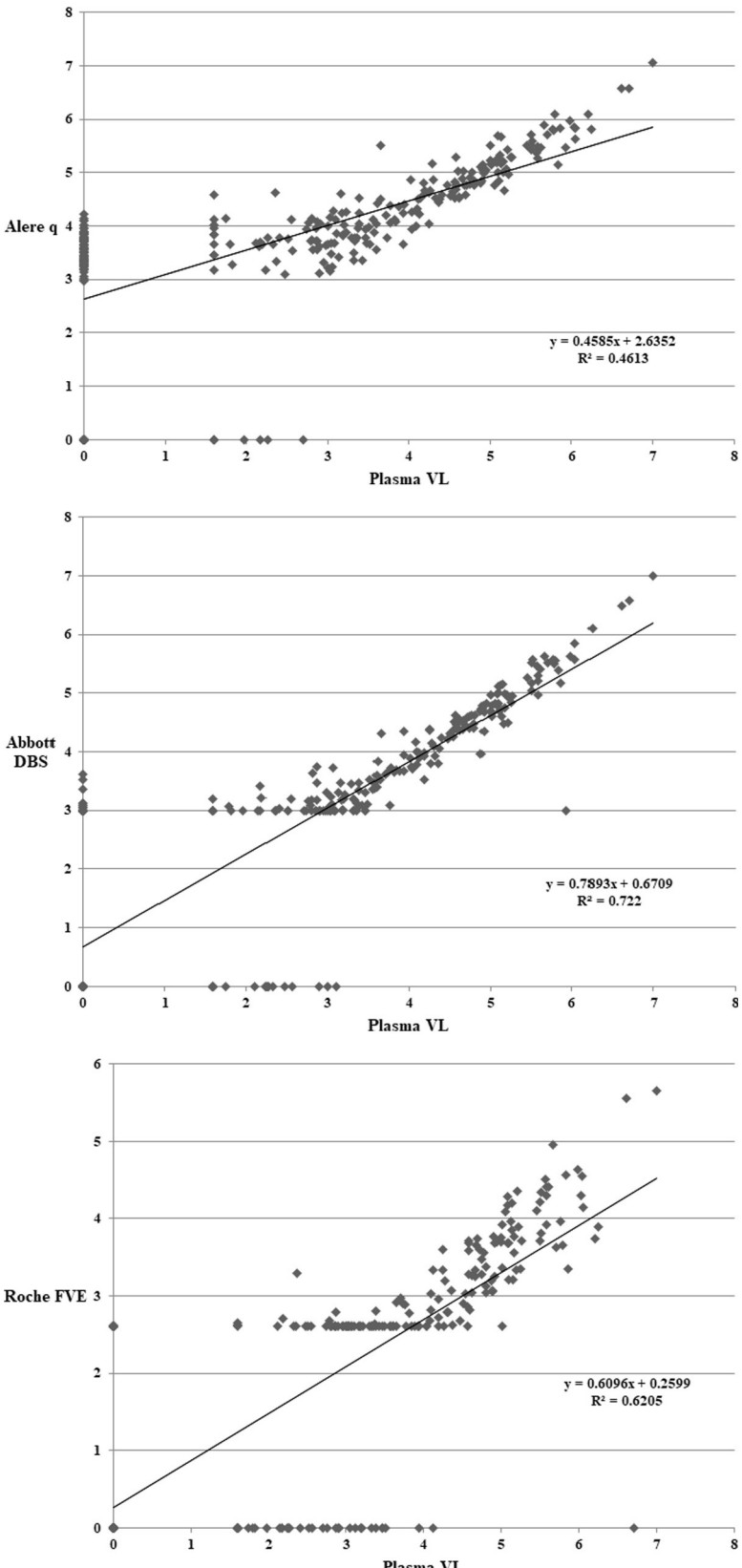

**Fig 1. Correlation plots between plasma VL and Alere q/DBS assays in $\log_{10}$ copies/ml.** *Correlation plots between plasma VL and 1a) Alere q—*$R^2$ = 0.46. *1b) Abbott DBS–*$R^2$ = 0.72. 2.9$\log_{10}$ has been used on plot for samples <1000 copies/ml (lower limit of quantification). *1c) Roche FVE—*$R^2$ = 0.62. 2.6$\log_{10}$ has been used on plot for Roche FVE samples <400 copies/ml (lower limit of quantification). Results below the detectable limit are plotted as 0 for all the plots.

## 2.3 Analysis

Sensitivity and specificity for identifying VF compared to the gold standard, plasma VL, was calculated. Plasma VL threshold of 1000 copies/ml were used as the gold standard for determining VF as this is the threshold recommended by WHO that is associated with treatment failure. Linear correlation and Bland-Altman plots were used to compare VL methodologies from a quantitative perspective using Microsoft Excel 2016 (Microsoft, USA), while sensitivity, specificity and correct classification of VF were calculated using Stata version 10 (StataCorp, USA). VL results are reported in copies/ml or $\log_{10}$ transformed copies/ml.

# 3. Results

## 3.1 Samples and patient characteristics

Two hundred and ninety-nine patient samples were selected for the study. The patients in each plasma VL categories were as follows: 94 (32%) were target not detected, 52 (17%) between <40–1000 copies/ml, 52 (17%) 1000–10,000 copies/ml, 53 (18%) 10,000–100,000 copies/ml and 48 (16%) >100,000 copies/ml. The median age of the patients was 35 years (interquartile range 28–41) with majority (93%) of samples from adults.

## 3.2 Overall performance and classification of VF

**Sensitivity** for identifying VF when plasma VL>1000 copies/ml: Alere q had the best sensitivity of 100% (95% CI 97–100%), followed by 91% (95% CI 85–95%) with Abbott DBS, while Roche FVE had a reduced sensitivity of 52% (95% CI 44–61%). Roche FVE showed 0% sensitivity when plasma VL was 1000–10,000 copies/ml but had 80% sensitivity when plasma VL was >10,000 copies/ml.

**Specificity** for identifying absence of VF when plasma VL<1000 copies/ml: Roche FVE had the best specificity of 99% (95% CI 95–100%), followed by 84% (95% CI 77–89%) with Abbott DBS, while Alere q had a reduced specificity of 19% (95% CI 13–27%).

Overall Alere q, Abbott DBS and Roche FVE correctly classified VF in 61%, 87% and 76% of samples respectively. Abbott DBS had the best correlation with plasma, followed by Roche FVE and Alere q (Figs 1 and 2). The performance varied across plasma VL categories (Table 2 and Figs 1 and 2). The greatest variability in classifying VF was observed in the <40–1000 copies/ml category.

# 4. Discussion

Our study compared several different approaches adopted by manufacturers in an attempt to provide alternatives to plasma VL testing. These methodologies differ in terms of sample treatment, nucleic acid extraction, amplification targets, detection methods and software cutoff algorithms, and thus the cause of the varied performances is likely multi-factorial.

Our study is one of the preliminary studies evaluating Alere q whole blood VL. Its advantages are POC test, ease of use, minimal training required, small portable instrument, small sample volume, built in controls, results within an hour and excellent sensitivity. These

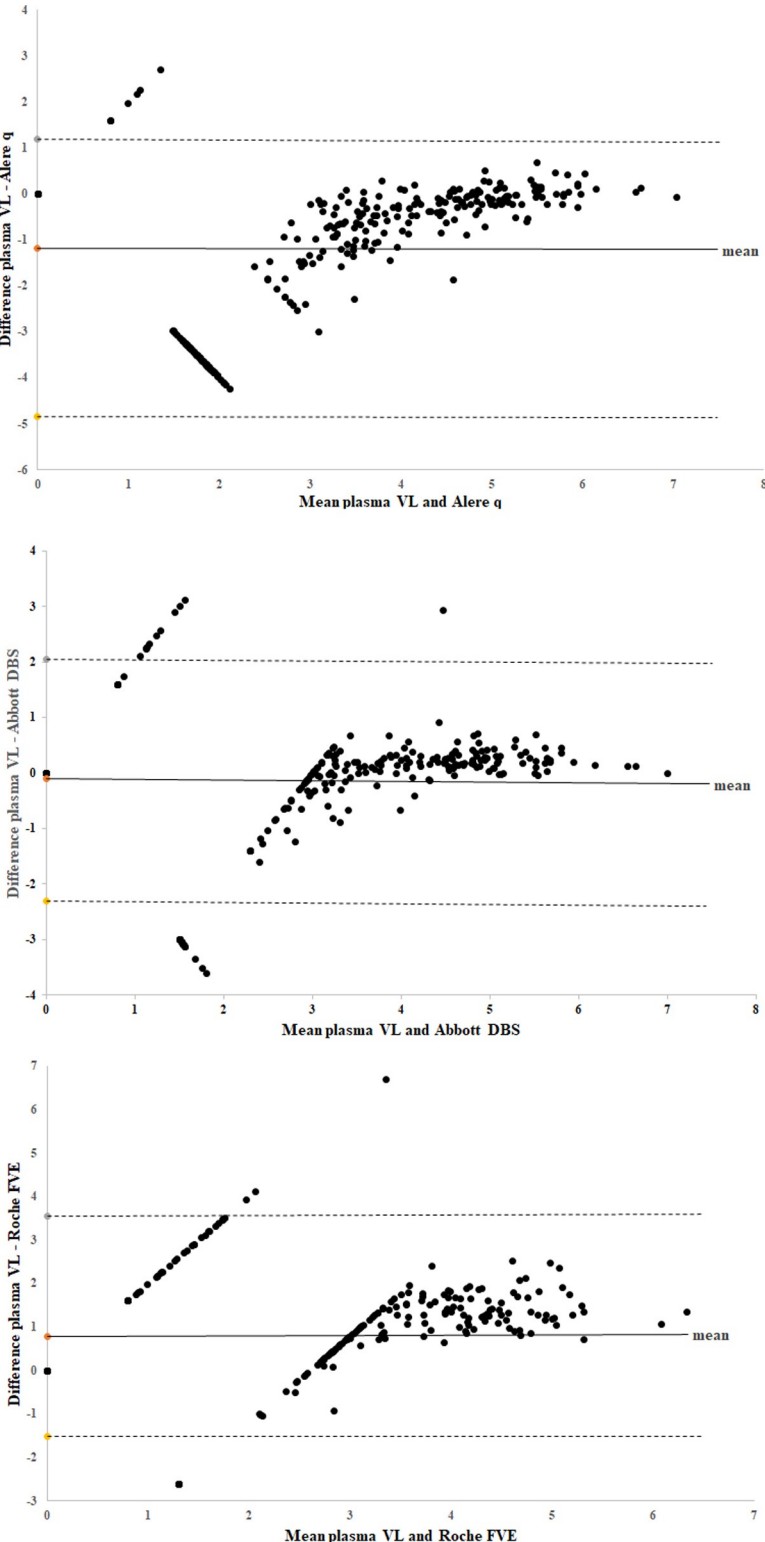

**Fig 2. Bland Altman plots between plasma VL and Alere q/DBS assays in log$_{10}$ copies/ml.** *Overall mean log$_{10}$ copies/ml difference between plasma VL and 2a) Alere q = -1.18 2b) Abbott DBS = -0.11 2c) Roche FVE = 0.78. Viral load that was below the detectable limit was plotted as 0 on graph, <1000 copies/ml for Abbott DBS was plotted as 2.9 log$_{10}$, and <400 copies/ml for Roche FVE was plotted as 2.6 log$_{10}$. The horizontal lines represent the mean (solid line) and the ± 1.96 standard deviations (dashed lines).*

**Table 2. Percentage of correct classification of VF by Alere q and DBS methods according to plasma VL categories.**

| Plasma VL categories | Target not detected | <40–1000 copies/ml | Specificity | 1000–10,000 copies/ml | >10,000 copies/ml | Sensitivity |
|---|---|---|---|---|---|---|
| Alere q | 22% (21/94) | 14% (7/51) | 19% (28/145) (95% CI 13–27%) | 100% (52/52) | 100% (101/101) | 100% (153/153) (95% CI 97–100%) |
| Abbott DBS | 89% (84/94) | 75% (38/51) | 84% (122/145) (95% CI 77–89%) | 75% (39/52) | 99% (97/98) | 91% (136/150) (95% CI 85–95%) |
| Roche FVE | 100% (90/90) | 98% (49/50) | 99% (139/140) (95% CI 95–100%) | 0% (0/50) | 80% (76/95) | 52% (76/145) (95% CI 44–61%) |

Definition of correct classification of VF by Alere q and DBS methods according to plasma VL categories using the VF threshold of 1000 copies/ml: when plasma VL < 1000 copies/ml (both target not detected and <40-1000copies/ml categories) then Alere q/DBS VL <1000 copies/ml (specificity in identifying absence of VF), when plasma VL > = 1000 copies/ml (both 1000–10,000 and >10,000 copies/ml categories) then Alere q/DBS VL > = 1000 copies/ml (sensitivity in identifying VF). Alere q showed 100% sensitivity, however there was reduced specificity in both the target not detected and <40–1000 copies/ml plasma VL categories.(22% and 14% respectively). For Abbott DBS, sensitivity was best above 10,000 copies/ml, however sensitivity was 75% in the 1000–10,000 copies/ml category, and specificity was 75% in the <40–1000 copies/ml category. Roche FVE demonstrated the opposite performance with 0% sensitivity in the 1000–10,000 copies/ml category but had the best specificity. Correct classification was best for all methods at >10,000 copies/ml.

advantages were confirmed in early infant diagnosis field study in South Africa where good performance was shown and use reported as highly acceptable [20].

Both Alere q whole blood VL and Abbott DBS measures cell associated HIV RNA as both extraction methods are RNA selective. The over-quantification was significantly greater with Alere q, probably due to other differences such as assay target, detection methodology, in conjunction with cell associated HIV RNA. However, the inclusion of cell-associated HIV RNA can vary greatly from one clinical sample to the next depending on clinical factors such as duration of viral suppression, CD4 count, haematocrit and other factors such as HIV DNA copy number in cells, as well as presence of spliced and unspliced HIV RNA in cells (low levels of unspliced RNA in patients suppressed on ART) [10, 21].

A study on Alere q using finger-prick capillary whole blood samples in a primary health care centre in Mozambique had similar results to ours albeit with better specificity, sensitivity was 96.83% and specificity was 47.8% [22]. Jani and colleagues suggested that a raised cut-off of 10,000 copies/ml would provide a better predictor of VF for Alere q and our independent ROC analysis confirms this finding [22]. Its low throughput and reduced specificity warrant further investigation to identify its niche use in large ART programmes. In a programmatic setting, individuals with VL's within this range would either require confirmatory plasma VL testing at a reference laboratory, or have further POC VL's to monitor the whole blood VL trajectory before a clinical decision can be made. Both of these approaches would result in delay and thus needs to be weighed against the risk of misclassification. The WHO 2016 consolidated guidelines on the use of antiretroviral drugs for treating and preventing HIV infection recommend the threshold of 1000 copies/ml for VF using DBS as well as a sensitivity and specificity >85%, therefore Alere q using whole blood does not fulfil this acceptability criteria.

Although the limit of detection of Alere q using whole blood is 2491 copies/ml, patients with VF have a plasma VL >1000 copies/ml and use of whole blood would include measurement of cell associated HIV RNA therefore elevating the viral load. Our analysis showed 100% sensitivity in detection of VF using Alere q including plasma VL's between 1000–3000 copies/ml.

Although DBS methodologies compares favourably in overall proportion of correctly identified VF, their lack of sensitivity and the variability between methodologies detected in our data is a concern. Reduced sensitivity of DBS is attributed to smaller volume of blood processed, reduced extraction efficiency and presence of PCR inhibitors [21, 23]. The DBS

protocols had the tendency to lower VL quantification when the plasma VL was 1000–10,000 copies/ml, as demonstrated by the reduced sensitivity in this category, as compared to a good sensitivity seen at >10,000 copies/ml plasma VL. The reduced sensitivity of Roche FVE, in particular, may be related to the inefficiency of the free virus elution and the lower input volume of DBS. Increasing the software correction factor as shown by Pollack et al. or using two or more spots may help to improve the sensitivity [24]. Pooled estimates of sensitivity and specificity based on a technical evaluation meta-analysis for Roche FVE was 95% sensitivity and 94% specificity while for Abbott DBS one spot was 88% sensitivity and 99% specificity [25]. Our study found a lower sensitivity (52%) with use of Roche FVE but specificity was comparable [19, 26, 27]. An evaluation of Roche FVE by Taieb et al. reported 55% sensitivity and 100% specificity, similar to our findings [28]. The sensitivity and specificity of Abbott DBS in our study is comparable to other studies evaluating the one spot or two spot protocol [14, 29–32].

A study evaluating DBS for VF by Sawadogo and colleagues concluded that the recommendations should not treat different assays and extraction methods as homogeneous and explicit recommendations for each should be available [33]. Inzaule and colleagues argue that the DBS cut-off of 1000 copies/ml is too stringent for VF based on their analysis of 79,566 VL results from patients receiving ART in western Kenya and will result in unnecessary repeat tests and regimen switches [34]. These studies, in conjunction with our finding of great variability between methodologies, suggest either some form of harmonization of the whole blood and DBS VL methodologies and interpretation is required, or other alternatives should be explored. In particular, the use of field plasma separation device and plasma separation cards could be a preferred method for performing VL in remote settings [25, 35].

Our study has the following limitations. Most significantly, DBS preparation and POC testing was performed in the laboratory in a controlled environment by skilled laboratory workers and not by the intended POC health care workers, as a result venous whole blood samples and not capillary blood was used for the POC and DBS assays in this evaluation. However, study that is conducted in the field would introduce even more variance and likely amplify the discrepancies between various technologies that we have found. The cross-sectional nature of the study precludes any follow-up analysis and with limited clinical and laboratory parameters, it is difficult gain insight into the exact cause of the inter-assay VL variance. Appropriately powered longitudinal studies would be required to address this issue.

## 5. Conclusion

Although various factors lead to the variability in whole blood and DBS VL quantification, the crucial advantage advocating their use is increasing access to VL. The strength of Alere q is its use as a POC test using whole blood and the excellent sensitivity of 100%, however this was offset by the specificity of only 19%. The over estimation of VL would lead to unnecessary re-testing and switching regimens. Optimisation of Alere q whole blood VL is required to facilitate the implementation of a cut-off with optimal sensitivity and specificity for VF. Both the Abbott DBS and Roche FVE protocols showed good specificity, however sensitivity was reduced when the plasma VL was 1000–10,000 copies/ml. This could result in delays in detecting VF and accumulation of drug resistance. Field evaluation in settings that have adopted these DBS protocols are necessary to gauge its real world performance.

## Supporting information

**S1 File. Receiver operating characteristic (ROC) curve analysis.**
(DOCX)

## Acknowledgments

The authors would like to thank Dr Sergio Carmona for the inter-laboratory collaboration and the staff at National Health Laboratory Service in Groote Schuur Hospital and Charlotte Maxeke Johannesburg Academic Hospital who assisted with the laboratory work.

Poster presentation of initial study findings at 8th IAS Conference on HIV Pathogenesis, Treatment & Prevention, 19–22 July 2015, Vancouver, Canada

## Author Contributions

**Conceptualization:** Nei-yuan Hsiao.

**Data curation:** Aabida Khan.

**Formal analysis:** Aabida Khan.

**Investigation:** Aabida Khan, Lucia Hans.

**Methodology:** Nei-yuan Hsiao.

**Supervision:** Nei-yuan Hsiao.

**Writing – original draft:** Aabida Khan.

**Writing – review & editing:** Aabida Khan, Lucia Hans, Nei-yuan Hsiao.

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
