## [Decision Letter · Decision Letter 0]

18 Dec 2019

PONE-D-19-29100

Comparison of Alere q whole blood viral load with DBS and plasma viral load in the classification of HIV virological failure

PLOS ONE

Dear Dr Khan,

Thank you for submitting your manuscript to PLOS ONE. After careful consideration, we feel that it has merit but does not fully meet PLOS ONE’s publication criteria as it currently stands. Therefore, we invite you to submit a revised version of the manuscript that addresses the points raised during the review process.

Two reviewers have raised serious issues with the manuscript as written, and each of their concerns must be addressed. Careful attention should be paid to use of "specificity" and "sensitivity" in the correct context, and provide sufficient detail of how the samples were created and how they were tested.

We would appreciate receiving your revised manuscript by Feb 01 2020 11:59PM. To enhance the reproducibility of your results, we recommend that if applicable you deposit your laboratory protocols in protocols.io, where a protocol can be assigned its own identifier (DOI) such that it can be cited independently in the future. For instructions see: http://journals.plos.org/plosone/s/submission-guidelines#loc-laboratory-protocols

We look forward to receiving your revised manuscript.

Kind regards,

Julie AE Nelson, PhD

Academic Editor

PLOS ONE

Journal Requirements:

1.

2. In ethics statement in the manuscript and in the online submission form, please provide additional information about the patient records/samples used in your retrospective study. Specifically, please ensure that you have discussed whether all data/samples were fully anonymized before you accessed them and/or whether the IRB or ethics committee waived the requirement for informed consent. If patients provided informed written consent to have data/samples from their medical records used in research, please include this information.

Reviewers' comments:

Reviewer's Responses to Questions

**Comments to the Author**

1. Is the manuscript technically sound, and do the data support the conclusions?

Reviewer #1: Partly

Reviewer #2: No

2. Has the statistical analysis been performed appropriately and rigorously? 

Reviewer #1: No

Reviewer #2: No

3. Have the authors made all data underlying the findings in their manuscript fully available?

Reviewer #1: Yes

Reviewer #2: No

4. Is the manuscript presented in an intelligible fashion and written in standard English?

Reviewer #1: Yes

Reviewer #2: Yes

5. Review Comments to the Author

Reviewer #1: The manuscript by Khan and colleagues presents a laboratory evaluation of the new Alere q point of care (POC) machine to measure HIV viral load from whole blood as compared to plasma which is the gold standard; interestingly, they also compare its performances to those of DBS, again as compared to plasma. This kind of evaluation is essential to help scaling-up of HIV viral load monitoring in resource-limited settings which still is scarce in many settings. In that respect, I am favorable to accept this manuscript if the modifications are conducted.

1) In the method section, the authors mention that the samples originate from routine care. The plasma is intended for HIV viral load monitoring, while the whole blood sample was intended for CD4 viral load measurement. Due to the sampling strategy, HIV viral load on plasma needed to be performed first before DBS samples were prepared and before a sample dedicated to Alere q was saved. Some details about the delay between whole blood sampling and DBS preparation, about the condition of storage of the whole blood, and about the condition and duration of storage of DBS would be welcome. Could the authors also mention if the CD4 measurement was performed or not.

2) In table 1, the reportable HIV viral load range of each technique is indicated. It is surprising to see that the range is 2491 copies/mL to 10 million copies/mL for the Alere q POC, and 1000 copies/mL to 10 million copies/mL for the Abbott DBS technique. This is not correct for the Abbott DBS technique. Moreover, this does not make sense given the evaluation they conduct to identify virological failure at the threshold of 1000 copies/mL. It is also surprising to read that each spot is 75µL as the manufacturers recommend 70µL/spot; 70µL is also mentioned in the reference 14 chosen by the authors in table 1, I was unable to identify reference 13. Tang et al. in Journal of clinical virology 2017 is probably a better reference.

3) In the “analysis” subsection of the “methods” section, it should be clearly stated that correlations and Bland-Altman analysis aim at comparing the methodologies from a quantitative perspective. The ROC curve analysis is mentioned in this section but never presented in the result section and briefly mentioned in the discussion; this is however not of prime interest in this study. If the authors insisted to present the results from the ROC curve, it should come after mentioning that the main goal of the evaluation is a qualitative evaluation, and estimates the ability of the different techniques to identify patients in virological failure defined at the threshold of 1000 copies/mL, when this threshold is used for both the technique evaluated and the plasma. This evaluation relies on the estimation of sensitivity and specificity of each technique as compared to plasma. These terms never appear in the method section while they are much more important than the “proportions of correctly classified”; indeed, the WHO guidelines state that DBS techniques should present a sensitivity and specificity >85% in order to be an acceptable option for HIV viral load monitoring, it is clearly not a criteria based simply on the proportion of correctly classified. The “analysis” subsection must be written in a more intelligible way, and the “results” section should clearly state the sensitivities and specificities as this is not done.

4) In the “results” section, nearly all the data from the table 2 are already in the text. Consider removing the table. Figure 1 provides no informative data; consider removing. The section 3.2 alternates between quantitative comparisons and qualitative comparisons. This should be rewritten for better clarity.

5) In the “results” section, the authors mention indifferently specificity or upward misclassification on the one hand, and sensitivity or downward misclassification on the other hand; this makes the reading very complex and confusing. The standard statistical method to compare a test to a gold standard relies on estimation of sensitivity and specificity. The authors should stick to these terms for the sake of comparison. In table 3, it would be very valuable to present not only % but numbers as well. Two columns could also be added to present sensitivity and specificity with their confidence intervals. This section suffers from repetition.

6) The Bland-Altman is supposed to be a representation of the mean of two techniques on the x-axis against the difference between the two techniques on the y-axis. The caption on the x-axis seem to indicate that the Bland-Altman analysis is not correctly conducted. A legend for the x-axis should be reported. Moreover, this analysis requires to represent three horizontal lines: the mean difference and the 95% confidence interval of the difference. None of these appear in the figures. This must be addressed as it is difficult with the current figures to judge the quality of the techniques.

7) In the discussion, the authors mention the DBS protocols had the tendency to lower viral load quantification. I disagree with this statement. It might be very true for the Roche protocol with a sensitivity to detect failure of 76%; this is also obvious on the scatter plot (especially if the authors add the line with equation y=x). This is less true with the Abbott protocol with sensitivity and specificity of 95% and 92%, respectively. The under-estimation of the viral load level with the Roche technique has already been described in the literature, and this should be reported here. The low sensitivity of the Roche DBS technique is particularly worrying as about 25% of those in failure would not be identified as such. The authors state that Alere q has a very good sensitivity, which is indeed 100%. However, the specificity was particularly bad (61%) which should be discussed. It looks like they attempt to discuss it by mentioning that in case of “false virological failure”, the patients would undergo confirmatory plasma VL testing at a laboratory of further POC VL monitoring. But this is not totally clear. The discussion needs clarification and should be put in perspectives with other similar studies as well as with the WHO recommendations.

Minor comments:

Throughout the manuscript, ul is written instead of µL.

IQR should be written in full at the first appearance

ART should be written in full at the first appearance, and not at its second appearance

Mention of “Alere q” is not homogeneous throughout the manuscript

Reviewer #2: The present study aims to address a cotemporary, globally important issue for HIV treatment and care; however, the manuscript in its current format needs significant improvement in order to reach publication standard. Following are some comments hopefully can help

Introduction

- The argument that “quality of evidence is low due to insufficient published data….” (Lines 45-47 page 3) is not well supported by existing evidence: According to a recent WHO report there were 40 technical evaluations of DBS across 25 countries examining 6 different commercially available VL technologies resulted in more than 10,000 paired DBS-plasma points. It suggested “sufficient evidence has been generated on the performance of DBS specimens for viral load testing to support the initiation of scale-up” (HIV molecular diagnostics toolkit to improve access to viral load testing and infant diagnostics, July 2019). The rationale for the present study need to be revised and strengthen

- Detailed descriptions of the POC VL test under investigation are needed. The Alere q POC viral load had been on the market (?) for a while (previous study published in 2016 – reference 19) why there is still “prototype” whole blood protocol? More details on operational requirements of the technologies are also needed

- Clear statement of the study objectives: primary and secondary objectives are recommended: which was the index test, which was the reference, and comparators

Methods

- Lines 60-61: … Lower than detection limit (LDL) of which technology? There were 3 different tech/protocols mentioned. Please explain why these specific technologies/protocol were selected? Was Sample size calculation performed?

- As this is a diagnostic accuracy study (as it appears in the introduction of the manuscript) , I’m not quite sure why was it needed/authors wanted) to collect data to “determine potential predictors of VL method discrepancies” ??? How did it come even before knowing there were “discrepancies” across measures in this study? As paired samples were used/tested

- VL threshold of 1000 copies/ml was used so detailed analysis for diagnostic accuracy of this cut-off needed to be described (sensitivity, specificity…). Same as for Bland-Altman analysis for quantitative measures

Results

- Lines 98-100 page 5 CD4 count, ART regimen/history not directly relevant/necessary for the main purpose of this study

- Lines 105-118 page 6: presentation of the results is not clear and somewhat repetitive. Suggest to separate out caparisons for primary and secondary objectives Alere q vs. plasma and Alere q vs DBS

- Lines 121-133 page 7 repeated results (with different expressions). Lines 146-154 page 8 not necessary/relevant

Discussion

- Lines 159-161 “the tendency of Alere q… most likely due to cell associated HIV RNA…” this is also true for DBS (which is also dried whole blood sample). Should/can the Alere q run on fresh plasma sample processing at site using mini centrifuge?

- Discussions of clinical factors influencing the variability in VL measures across technologies are not necessary. Potential reasons for reduced accuracy of Roche DBS/FVE has been extensively discussed in literature

- The recommendation of increasing correction factor or using more spots are too general and prompt the question of why this challenge has not been addressed even though is has been identified previously (low specificity)?

- The “excellent sensitivity” note (of Alere q whole blood) is somewhat misleading without an equal highlighted note/discussion of “low specificity” of <50%, which is consistent with previous research and suggested that the technology has not been improved over last few years??? This has an important implication for field implementation as it indicates very high/unacceptable over estimation of HIV VL leading to unnecessary retesting/confirmatory testing of treatment failure and switching regimen

- Lines 199-200 please check the references

Conclusion

- Not clear on key messages. Again highlighting the “excellent sensitivity” without noting low specificity is misleading and based on results of this study the use of “prototype” whole blood protocol Alere q for HIV VL testing should not be “warranted” without further technology improvement and validation

Figures 3a/b BA plots should be revised clearly shown 95%CI of the mean deferences & LoAs

6. PLOS authors have the option to publish the peer review history of their article (what does this mean?). If published, this will include your full peer review and any attached files.

Reviewer #1: No

Reviewer #2: Yes: Minh D Pham

---

## [Author Response · Author response to Decision Letter 0]

7 Feb 2020

5. Review Comments to the Author

Reviewer #1: The manuscript by Khan and colleagues presents a laboratory evaluation of the new Alere q point of care (POC) machine to measure HIV viral load from whole blood as compared to plasma which is the gold standard; interestingly, they also compare its performances to those of DBS, again as compared to plasma. This kind of evaluation is essential to help scaling-up of HIV viral load monitoring in resource-limited settings which still is scarce in many settings. In that respect, I am favorable to accept this manuscript if the modifications are conducted.

1) In the method section, the authors mention that the samples originate from routine care. The plasma is intended for HIV viral load monitoring, while the whole blood sample was intended for CD4 viral load measurement. Due to the sampling strategy, HIV viral load on plasma needed to be performed first before DBS samples were prepared and before a sample dedicated to Alere q was saved. Some details about the delay between whole blood sampling and DBS preparation, about the condition of storage of the whole blood, and about the condition and duration of storage of DBS would be welcome. Could the authors also mention if the CD4 measurement was performed or not.

Answer: To ensure most of the VL analytical spectrum was covered, our convenient sampling strategy randomly selected paired CD4/VL EDTA samples where at least 50 samples of plasma VL results were from each of four categories: <40 copies/ml, 40–1000 copies/ml, 1000 – 10,000 copies/ml and > 10,000 copies/ml. Once a plasma VL and CD4 testing were complete, the remaining paired CD4 sample was used for whole blood Alere q test and DBS within 72 hours of sample receipt to prevent sample degradation. DBS was prepared by applying 75µL of whole blood to each spot on a Whatman 903 filter-paper card, dried and packed into plastic bags with desiccant sachets and stored at room temperature in boxes away from direct sunlight or heat. DBS VLs were tested in batches within 1-2 months of preparation. Whole blood specimen that was used for the DBS and Alere q testing was stored at room temperature after CD4 testing. 

2) In table 1, the reportable HIV viral load range of each technique is indicated. It is surprising to see that the range is 2491 copies/mL to 10 million copies/mL for the Alere q POC, and 1000 copies/mL to 10 million copies/mL for the Abbott DBS technique. This is not correct for the Abbott DBS technique. Moreover, this does not make sense given the evaluation they conduct to identify virological failure at the threshold of 1000 copies/mL. It is also surprising to read that each spot is 75µL as the manufacturers recommend 70µL/spot; 70µL is also mentioned in the reference 14 chosen by the authors in table 1, I was unable to identify reference 13. Tang et al. in Journal of clinical virology 2017 is probably a better reference.

Answer: Alere q requires 25µL whole blood, due to the smaller sample volume as compared to DBS and plasma the limit of detection is therefore lower. According to the package insert and the WHO Prequalification of In Vitro Diagnostics public report on Alere™ q HIV-1/2 Detect, which was designed as a qualitative nucleic acid amplification test for the detection of Human Immunodeficiency Virus (HIV) type 1 groups M/N and O and type 2, the limit of detection for HIV-1 group M is 2491 copies/ml. (Available from: https://www.who.int/diagnostics_laboratory/evaluations/pq-list/hiv-vrl/160613PQPublicReport_0226-032-00AlereHIVDetect_v2.pdf.). Patients with virological failure have a plasma VL >1000 copies/ml. Use of whole blood would include measurement of cell associated HIV RNA therefore elevating the VL. Our analysis showed 100% sensitivity in detection of virological failure using Alere q including plasma VL’s between 1000-3000 copies/ml.

In reference 17, “Smit PW, Sollis KA, Fiscus S, Ford N, Vitoria M, Essajee S, et al. Systematic review of the use of dried blood spots for monitoring HIV viral load and for early infant diagnosis. PloS one. 2014;9(3):e86461” – thirteen studies evaluated DBS HIV viral load and the volume of blood per DBS spot ranged from 50–100µL. Most manufacturers do recommend 70µL/spot. To optimize sensitivity and facilitate full coverage of spot, 75µL of whole blood was applied to each spot. 

The Abbott DBS protocol that was evaluated is a revised prototype protocol that uses one DBS spot (reportable range 1000-10 million copies/ml) as compared to two DBS spots used by the Abbott RealTime HIV-1 original ‘open mode’ protocol (reportable range 550-10 million copies/ml). Reference 13, “Shihai Huang SC, Brian Erickson, John Salituro, Chadwick Dunn, Livhuwani Nxumalo, Jeffrey Wuitschick, Jens Dhein. Improved Performance of the New Prototype Automated Abbott RealTime HIV-1 DBS Viral Load Assay: Potential Use in Expanded Viral Load Testing in Resource Limited Setting. ASLM2014” was presented as an oral and poster presentation at the African Society for Laboratory Medicine in 2014(From: https://www.eiseverywhere.com/docs/4466/ScientificProgrammeBook). There are several workflow improvements with the revised prototype protocol as compared to the original ‘open mode’ protocol which includes use of one DBS spot, number of sample tubes reduced from two to one, automated DBS eluate transfer and up to 93 DBS specimens per run therefore being more cost effective. We had also evaluated the original ‘open mode’ protocol however chose not to include the data as the results from both protocols were similar.

Have added the Tang et al. in Journal of clinical virology 2017, as it also evaluates the revised Abbott DBS protocol and is more recent.

3) In the “analysis” subsection of the “methods” section, it should be clearly stated that correlations and Bland-Altman analysis aim at comparing the methodologies from a quantitative perspective. The ROC curve analysis is mentioned in this section but never presented in the result section and briefly mentioned in the discussion; this is however not of prime interest in this study. If the authors insisted to present the results from the ROC curve, it should come after mentioning that the main goal of the evaluation is a qualitative evaluation, and estimates the ability of the different techniques to identify patients in virological failure defined at the threshold of 1000 copies/mL, when this threshold is used for both the technique evaluated and the plasma. This evaluation relies on the estimation of sensitivity and specificity of each technique as compared to plasma. These terms never appear in the method section while they are much more important than the “proportions of correctly classified”; indeed, the WHO guidelines state that DBS techniques should present a sensitivity and specificity >85% in order to be an acceptable option for HIV viral load monitoring, it is clearly not a criteria based simply on the proportion of correctly classified. The “analysis” subsection must be written in a more intelligible way, and the “results” section should clearly state the sensitivities and specificities as this is not done.

Answer: See revised manuscript for re-written analysis section of methods as per reviewer’s comments.

4) In the “results” section, nearly all the data from the table 2 are already in the text. Consider removing the table. Figure 1 provides no informative data; consider removing. The section 3.2 alternates between quantitative comparisons and qualitative comparisons. This should be rewritten for better clarity.

Answer: Removed table 2 and figure 1 as per reviewer’s recommendation. Re-written section 3.2.

5) In the “results” section, the authors mention indifferently specificity or upward misclassification on the one hand, and sensitivity or downward misclassification on the other hand; this makes the reading very complex and confusing. The standard statistical method to compare a test to a gold standard relies on estimation of sensitivity and specificity. The authors should stick to these terms for the sake of comparison. In table 3, it would be very valuable to present not only % but numbers as well. Two columns could also be added to present sensitivity and specificity with their confidence intervals. This section suffers from repetition.

Answer: To use the terms sensitivity and specificity as recommended and remove upward/downward misclassification. Table amended as per reviewer recommendations.

6) The Bland-Altman is supposed to be a representation of the mean of two techniques on the x-axis against the difference between the two techniques on the y-axis. The caption on the x-axis seem to indicate that the Bland-Altman analysis is not correctly conducted. A legend for the x-axis should be reported. Moreover, this analysis requires to represent three horizontal lines: the mean difference and the 95% confidence interval of the difference. None of these appear in the figures. This must be addressed as it is difficult with the current figures to judge the quality of the techniques.

Answer: See revised Bland-Altman analysis as per reviewer recommendations.

7) In the discussion, the authors mention the DBS protocols had the tendency to lower viral load quantification. I disagree with this statement. It might be very true for the Roche protocol with a sensitivity to detect failure of 76%; this is also obvious on the scatter plot (especially if the authors add the line with equation y=x). This is less true with the Abbott protocol with sensitivity and specificity of 95% and 92%, respectively. The under-estimation of the viral load level with the Roche technique has already been described in the literature, and this should be reported here. The low sensitivity of the Roche DBS technique is particularly worrying as about 25% of those in failure would not be identified as such. The authors state that Alere q has a very good sensitivity, which is indeed 100%. However, the specificity was particularly bad (61%) which should be discussed. It looks like they attempt to discuss it by mentioning that in case of “false virological failure”, the patients would undergo confirmatory plasma VL testing at a laboratory of further POC VL monitoring. But this is not totally clear. The discussion needs clarification and should be put in perspectives with other similar studies as well as with the WHO recommendations.

Answer: Performance of DBS VL is highly variable. The DBS protocols had the tendency to lower viral load quantification when the plasma VL was 1000-10,000 copies/ml, as demonstrated by the reduced sensitivity in this category, as compared to a good sensitivity seen at >10,000 copies/ml plasma VL. Other studies have evaluated the Roche free virus elution (FVE) protocol. An evaluation by Wu et al. had a 90% sensitivity and 100% specificity from paired DBS and plasma samples from 196 patients, with plasma VLs ranging from undetectable to 106 copies/ml. Another study by Makadzange et al. had a 92.7% sensitivity and 100% specificity and found that DBS viral load values were 1 log10 copies/ml lower than those obtained by plasma. Pannus et al. evaluated both Roche FVE and Abbott DBS which had 80.8% and 76% sensitivity and 87.3% and 89.7% specificity respectively. Pooled estimates of sensitivity and specificity based on published data up to June 2015 in the WHO HIV treatment guidelines 2016 for Roche FVE was 85% sensitivity and 94% specificity while for Abbott DBS was 95% sensitivity and 92% specificity. Our study found a lower sensitivity (52%) with use of Roche FVE but specificity was comparable. The sensitivity and specificity of Abbott DBS in our study is comparable to other studies (Erba et al, Rutstein et al, Zeh et al.).

The reduced specificity of Alere q is most likely due to the inclusion of cell associated HIV RNA in its methodology. The inclusion of cell-associated HIV RNA can vary greatly from one clinical sample to the next depending on clinical factors such as duration of viral suppression, CD4 count, haematocrit and other factors such as HIV DNA copy number in cells, as well as presence of spliced and unspliced HIV RNA in cells. A study on Alere q POC VL using finger-prick capillary whole blood samples in a primary health care centre in Mozambique had similar results to ours albeit with better specificity, sensitivity was 96.83% and specificity was 47.8%. Jani and colleagues suggested that a raised cut-off of 10,000 copies/ml would provide a better predictor of VF for Alere q and our independent ROC analysis confirms this finding. However the WHO 2016 consolidated guidelines on the use of antiretroviral drugs for treating and preventing HIV infection recommend the threshold of 1000 copies/ml for virological failure using DBS as well as a sensitivity and specificity >85%.

Minor comments:

Throughout the manuscript, ul is written instead of µL. Answer: Changed to µL

IQR should be written in full at the first appearance Answer: Amended

ART should be written in full at the first appearance, and not at its second appearance Answer: Amended

Mention of “Alere q” is not homogeneous throughout the manuscript Answer: Amended

Reviewer #2: The present study aims to address a cotemporary, globally important issue for HIV treatment and care; however, the manuscript in its current format needs significant improvement in order to reach publication standard. Following are some comments hopefully can help

Introduction

- The argument that “quality of evidence is low due to insufficient published data….” (Lines 45-47 page 3) is not well supported by existing evidence: According to a recent WHO report there were 40 technical evaluations of DBS across 25 countries examining 6 different commercially available VL technologies resulted in more than 10,000 paired DBS-plasma points. It suggested “sufficient evidence has been generated on the performance of DBS specimens for viral load testing to support the initiation of scale-up” (HIV molecular diagnostics toolkit to improve access to viral load testing and infant diagnostics, July 2019). The rationale for the present study need to be revised and strengthen

Answer: The main rationale for this study was to evaluate the performance of a prototype Alere q HIV VL assay using whole blood and compare it against plasma VL and DBS VL in the classification of virological failure, as the need remains to evaluate POC technologies, to add to the existing DBS VL data due to the variability in testing protocols, differences in methodologies affecting sensitivity and specificity, the need for standardisation and the greater need for validated assays with regulatory approval. New POC technologies such as Alere q presents a potential alternative to the current plasma or DBS VL testing in specialised VL facilities. POC testing would reduce the need for expensive laboratory infrastructure and highly skilled laboratory workers, and also could lower the cost of testing. Rapid, reliable and affordable POC VL, with minimal equipment and training requirements, would have a major impact on treatment outcomes by facilitating timely detection of virological failure and swift clinical decision making as well as reduce loss to follow up.

The 2016 WHO consolidated guidelines on the use of antiretroviral drugs for treating and preventing HIV infection have recommended the use of DBS in settings where plasma VL cannot be done due to logistical, infrastructural or operational barriers. Only two commercial assays have received CE-IVD and WHO prequalification regulatory approval for viral load using DBS (Abbott RealTime HIV-1 and bioMérieux NucliSENS EasyQ® HIV-1). The systematic review and meta-analysis on the performance of using DBS for viral load testing which includes the 40 technical evaluations from 25 countries is currently unpublished. The sample size evaluated for Abbott DBS and Roche FVE was only 2704 and 3076 respectively. 

- Detailed descriptions of the POC VL test under investigation are needed. The Alere q POC viral load had been on the market (?) for a while (previous study published in 2016 – reference 19) why there is still “prototype” whole blood protocol? More details on operational requirements of the technologies are also needed

Answer: The prototype Alere q HIV-1/2 assay was evaluated in this study as specimens were tested in 2014 and 2015 using the prototype available for evaluation during this time period. The Alere™ q HIV-1/2 Detect has undergone WHO prequalification as a qualitative nucleic acid amplification test for the detection of HIV in plasma and whole blood in 2016. Abbott Diagnostics is now the supplier of Alere, and offer the viral load as the m-PIMA™ HIV-1/2 VL test which is commercially available, and has received CE-IVD marking and WHO prequalification using plasma in 2019. Its advantages are point of care test, ease of use, minimal training required, small portable instrument, small sample volume, built in controls, results within an hour and excellent sensitivity. The whole blood assay however is more advantageous as a point of care test as capillary whole blood can be collected from a finger/heel prick directly for testing. Operational requirements are minimal as compared to laboratory based assays as the cartridge kits can be kept at room temperature therefore no refrigeration is required. It can be used in a non-laboratory environment and can be operated on external power, with an internal rechargeable battery protecting it against power fluctuations. It does not require any user related instrument maintenance. 

- Clear statement of the study objectives: primary and secondary objectives are recommended: which was the index test, which was the reference, and comparators

Answer: Study objective

To evaluate the performance of a prototype Alere q whole blood VL POC assay in the classification of virological failure using plasma VL as the gold standard.

To evaluate the performance of locally available commercial VL assays (Abbott and Roche) using DBS protocols in the classification of virological failure using plasma VL as the gold standard.

Methods

- Lines 60-61: … Lower than detection limit (LDL) of which technology? There were 3 different tech/protocols mentioned. Please explain why these specific technologies/protocol were selected? Was Sample size calculation performed?

Answer: Plasma VL was used as the gold standard. Plasma specimens routinely tested by the Abbott RealTime HIV-1 assay were selected. To ensure most of the VL analytical spectrum was covered, our convenient sampling strategy randomly selected plasma VL results to include at least 50 specimens from each of four categories: <40 copies/ml, 40–1000 copies/ml, 1000 – 10,000 copies/ml and > 10,000 copies/ml. 

The prototype Alere q whole blood VL point of care assay was available for evaluation as an attractive means to increase access to VL testing. The DBS protocols were selected as Abbott and Roche plasma viral load platforms are locally available in our setting.

To ensure most of the VL analytical spectrum was covered, our convenient sampling strategy randomly selected plasma VL results to include at least 50 specimens from each of four categories listed above. Testing was then done based on the number of Alere q whole blood VL cartridges available, to include a total of 299 with results.

- As this is a diagnostic accuracy study (as it appears in the introduction of the manuscript) , I’m not quite sure why was it needed/authors wanted) to collect data to “determine potential predictors of VL method discrepancies” ??? How did it come even before knowing there were “discrepancies” across measures in this study? As paired samples were used/tested

Answer: It was anticipated from previous studies on DBS VL and WHO guidelines that discrepancies would occur as plasma measures cell free virus only while whole blood/DBS also includes cell associated virus. The inclusion of cell-associated HIV RNA can vary greatly from one clinical sample to the next depending on clinical factors such as duration of viral suppression, CD4 count, haematocrit and other factors such as HIV DNA copy number in cells, as well as presence of spliced and unspliced HIV RNA in cells (low levels of unspliced RNA in patients suppressed on ART). The volume of blood per DBS spot can range 50-100µL therefore can lead to reduced sensitivity as compared to plasma VL that tests 200-600µL. These factors were anticipated to impact on sensitivity and specificity.

- VL threshold of 1000 copies/ml was used so detailed analysis for diagnostic accuracy of this cut-off needed to be described (sensitivity, specificity…). Same as for Bland-Altman analysis for quantitative measures

Answer: See revised manuscript for sensitivity and specificity. See revised Bland-Altman analysis as per reviewer recommendations.

Results

- Lines 98-100 page 5 CD4 count, ART regimen/history not directly relevant/necessary for the main purpose of this study

Answer: Removed CD4 count and ART history.

- Lines 105-118 page 6: presentation of the results is not clear and somewhat repetitive. Suggest to separate out caparisons for primary and secondary objectives Alere q vs. plasma and Alere q vs DBS

Answer: See revised manuscript for amended results.

- Lines 121-133 page 7 repeated results (with different expressions). Lines 146-154 page 8 not necessary/relevant

Answer: Figure 1 removed. Table amended as per reviewer 1 recommendations. Can remove lines 146-154 page 8.

Discussion

- Lines 159-161 “the tendency of Alere q… most likely due to cell associated HIV RNA…” this is also true for DBS (which is also dried whole blood sample). Should/can the Alere q run on fresh plasma sample processing at site using mini centrifuge?

Answer: Yes, both Alere q whole blood assay and Abbott DBS protocols measures cell associated HIV RNA, both extraction methods are RNA selective. The over-quantification was significantly greater with Alere q, probably due to other differences such as assay target, detection methodology, in conjunction with cell associated HIV RNA. The Roche FVE measures only extracellular HIV RNA similar to plasma therefore it demonstrated the best specificity.

Abbott Diagnostics is now the supplier of Alere, and offer the viral load as the m-PIMA™ HIV-1/2 VL test which is commercially available, and has received CE-IVD marking and WHO prequalification using plasma in 2019. It requires the collection of an EDTA whole blood sample which requires centrifugation so that 50μL of plasma can be used for testing.

- Discussions of clinical factors influencing the variability in VL measures across technologies are not necessary. Potential reasons for reduced accuracy of Roche DBS/FVE has been extensively discussed in literature

Answer: Removed section 3.3.

- The recommendation of increasing correction factor or using more spots are too general and prompt the question of why this challenge has not been addressed even though is has been identified previously (low specificity)?

Answer: Pooled estimates of sensitivity and specificity based on published data up to June 2015 in the WHO HIV treatment guidelines 2016 for Roche FVE was 85% sensitivity and 94% specificity. Our study found a lower sensitivity (52%) with use of Roche FVE but specificity was comparable. The results from the technical evaluation meta-analysis in the WHO HIV molecular diagnostics toolkit to improve access to viral load testing and infant diagnostics 2019, showed a sensitivity of 95% and specificity of 94% for Roche FVE. This has shown a significant improvement since previous estimates. The meta-analysis is unpublished and there is no indication of a change in testing methodology or processing.

- The “excellent sensitivity” note (of Alere q whole blood) is somewhat misleading without an equal highlighted note/discussion of “low specificity” of <50%, which is consistent with previous research and suggested that the technology has not been improved over last few years??? This has an important implication for field implementation as it indicates very high/unacceptable over estimation of HIV VL leading to unnecessary retesting/confirmatory testing of treatment failure and switching regimen

Answer: Have amended discussion - Alere q whole blood assay had an excellent sensitivity of 100%, however this was offset by the specificity of only 19%. The over estimation of HIV VL would lead to unnecessary retesting/confirmatory testing of treatment failure and switching regimens. The WHO 2016 consolidated guidelines on the use of antiretroviral drugs for treating and preventing HIV infection recommend a sensitivity and specificity >85% using DBS for virological failure.

- Lines 199-200 please check the references

Answer: Checked reference 22 “Carmona S, Seiverth B, Magubane D, Hans L, Hoppler M. Separation of Plasma from Whole Blood by Use of the cobas Plasma Separation Card: a Compelling Alternative to Dried Blood Spots for Quantification of HIV-1 Viral Load. Journal of clinical microbiology. 2019;57(4):e01336-18” – evaluation of plasma separation card therefore kept. 

Removed reference 23 “Zeh C, Ndiege K, Inzaule S, Achieng R, Williamson J, Chang JC-W, et al. Evaluation of the performance of Abbott m2000 and Roche COBAS Ampliprep/COBAS Taqman assays for HIV-1 viral load determination using dried blood spots and dried plasma spots in Kenya. PloS one. 2017;12(6):e0179316” removed as plasma separation device/card not used to prepare dried plasma spot. Reference changed to WHO HIV molecular diagnostics toolkit to improve access to viral load testing and infant diagnostics July 2019 which includes use of plasma separation device/card.

Conclusion

- Not clear on key messages. Again highlighting the “excellent sensitivity” without noting low specificity is misleading and based on results of this study the use of “prototype” whole blood protocol Alere q for HIV VL testing should not be “warranted” without further technology improvement and validation

Answer: See revised manuscript for revised conclusion based on recommendations.

Figures 3a/b BA plots should be revised clearly shown 95%CI of the mean deferences & LoAs

Answer: See revised Bland-Altman analysis as per reviewer recommendations.

---

## [Decision Letter · Decision Letter 1]

25 Feb 2020

PONE-D-19-29100R1

Comparison of Alere q whole blood viral load with DBS and plasma viral load in the classification of HIV virological failure

PLOS ONE

Dear Dr Khan,

Thank you for submitting your manuscript to PLOS ONE. After careful consideration, we feel that it has merit but does not fully meet PLOS ONE’s publication criteria as it currently stands. Therefore, we invite you to submit a revised version of the manuscript that addresses the points raised during the review process.

Remaining issues to address include those of Reviewer 1 below. In particular, the study needs to include data from after 2015. While it is true that the data presented are from the earlier time, more studies have been published and the pooled results must be addressed to make this manuscript relevant to 2020. 

The authors must also address the following issues. The samples included in the testing were all from HIV-infected individuals, although some were not detectable by the Abbott gold standard assay. The Abbott assay has both <40cp/mL detected and <40cp/mL not detected as result options, so it must be addressed if these two categories were combined as not detected or if they were split, putting the detected ones with the 40-1000cp/mL group. Overall, there must be a better description of how the data were compared. There is a description of the VF level, but it is unclear how that was applied to the data from samples with <1000cp/mL as the plasma VL. The authors must be very specific: how were the data from each of the assays treated to decide if they had the correct classification? In particular, Table 2 is confusing for this reason: for the <40 sample category, was it that 22% were correctly not detected by Alere (and the others were incorrectly detected) or that 22% were correctly detected by Alere (and the other were incorrectly not detected)? For the other groups, what was considered "correct classification", detected at any number or detected at a number in the same range (such as 1000-10,000)? Or were they correctly classified if they were above 1000cp/mL as the VF threshold? There are no figure legends to help describe the figures, and there are no differences in markers between the groups to see how the data in the table fit with the graphs. The authors must add more transparency to their results and calculations to make their data useful to the wider research community.

We would appreciate receiving your revised manuscript by Apr 10 2020 11:59PM. To enhance the reproducibility of your results, we recommend that if applicable you deposit your laboratory protocols in protocols.io, where a protocol can be assigned its own identifier (DOI) such that it can be cited independently in the future. For instructions see: http://journals.plos.org/plosone/s/submission-guidelines#loc-laboratory-protocols

We look forward to receiving your revised manuscript.

Kind regards,

Julie AE Nelson, PhD

Academic Editor

PLOS ONE

Reviewers' comments:

Reviewer's Responses to Questions

**Comments to the Author**

1. If the authors have adequately addressed your comments raised in a previous round of review and you feel that this manuscript is now acceptable for publication, you may indicate that here to bypass the “Comments to the Author” section, enter your conflict of interest statement in the “Confidential to Editor” section, and submit your "Accept" recommendation.

Reviewer #1: (No Response)

2. Is the manuscript technically sound, and do the data support the conclusions?

Reviewer #1: Yes

3. Has the statistical analysis been performed appropriately and rigorously? 

Reviewer #1: Yes

4. Have the authors made all data underlying the findings in their manuscript fully available?

Reviewer #1: Yes

5. Is the manuscript presented in an intelligible fashion and written in standard English?

Reviewer #1: Yes

6. Review Comments to the Author

Reviewer #1: The authors have made great efforts to answer to the comments, and the article has improved considerably. I still have some minor comments however before the manuscript is deemed suitable for publication.

1. In the method section, the very first sentence is in contradiction with the following. Indeed, the authors state that the goals are to compare HIV viral load results obtained with Alere q POC with plasma HIV viral load results on the one hand, and with DBS viral load results on the other hand. However, two sentences later thy mention that secondary objectives are to compare DBS viral load results to plasma viral load results.

I guess the first sentence can be removed.

2. Line 73, when the authors say “the remaining paired CD4 sample was used for whole blood Alere q test…” I think that the terminlogy “CD4 sample was used” is not correct. Wouldn’t it be more accurate to say something like “the remainis of the blood sample for CD4 measurement”?

3. In current reference 14 (as cited in Table 1) , the limit of detection of the Abbott technique was found to be 839 copies/mL. This is also the threshold mentioned in the WHO pre-qualification document (1) This is also the technique specified in Taieb et al (PLOS one 2018) following the manufacturer’s information.

4. In the statistical method section, reference to ROC curve appears but no such result is presented. Consider removing the sentence referring to ROC curves.

5. In the results section, lines 140-144 are strangely placed. Consider moving them before the paragraph currently starting line 132.

6. In the discussion, I do not understand why the authors specify that they stopped their literature search in June 2015. This does not make sense in 2020. Moreover, some articles evaluating the DBS techniques of interest have been published since. These articles should also be regarded especially since some articles fou,d similar results with the Roche technique (e.g. Taieb et al. OFID 2016).

7. I still have issues with the Bland-Altman analysis. The figure should include the 95% confidence limits of the differences. I doubt the two horizontal grey lines correspond to these limits. The figures must be amended.

(1) https://www.who.int/diagnostics_laboratory/evaluations/pq-list/hiv-vrl/170830_amended_final_pr_0145_027_00.pdf

7. PLOS authors have the option to publish the peer review history of their article (what does this mean?). If published, this will include your full peer review and any attached files.

Reviewer #1: No

---

## [Author Response · Author response to Decision Letter 1]

6 Apr 2020

6. Review Comments to the Author

Reviewer #1: The authors have made great efforts to answer to the comments, and the article has improved considerably. I still have some minor comments however before the manuscript is deemed suitable for publication.

1. In the method section, the very first sentence is in contradiction with the following. Indeed, the authors state that the goals are to compare HIV viral load results obtained with Alere q POC with plasma HIV viral load results on the one hand, and with DBS viral load results on the other hand. However, two sentences later thy mention that secondary objectives are to compare DBS viral load results to plasma viral load results.

I guess the first sentence can be removed.

Answer: removed as per reviewer comment.

2. Line 73, when the authors say “the remaining paired CD4 sample was used for whole blood Alere q test…” I think that the terminlogy “CD4 sample was used” is not correct. Wouldn’t it be more accurate to say something like “the remainis of the blood sample for CD4 measurement”?

Answer: amended as per reviewer comment.

3. In current reference 14 (as cited in Table 1) , the limit of detection of the Abbott technique was found to be 839 copies/mL. This is also the threshold mentioned in the WHO pre-qualification document (1) This is also the technique specified in Taieb et al (PLOS one 2018) following the manufacturer’s information.

Answer: During the period of evaluation in 2014-2015, the Abbott RealTime HIV-1 DBS revised prototype protocol was under research use and not WHO pre-qualified. The limit of detection was reported at 1000 copies/ml (Shihai Huang SC, Brian Erickson, John Salituro, Chadwick Dunn, Livhuwani Nxumalo, Jeffrey Wuitschick, Jens Dhein. Improved Performance of the New Prototype Automated Abbott RealTime HIV-1 DBS Viral Load Assay: Potential Use in Expanded Viral Load Testing in Resource Limited Setting. ASLM2014) . Detectable results below the limit of detection were reported as <1000 copies/ml. WHO pre-qualification and CE marking of the one spot Abbott RealTime HIV-1 DBS protocol was done in 2016 in which the limit of detection was found to be 839 copies/ml (WHO prequalification of diagnostics programme public report 2016 and Tang N, Pahalawatta V, Frank A, Bagley Z, Viana R, Lampinen J, et al. HIV-1 viral load measurement in venous blood and fingerprick blood using Abbott RealTime HIV-1 DBS assay. Journal of Clinical Virology. 2017;92:56-61). Field evaluation by Taieb et al (PLOS one 2018) was done in 2017. Will remove reference Tang et al from Table 1 and discuss it under section 2.2 laboratory testing.

4. In the statistical method section, reference to ROC curve appears but no such result is presented. Consider removing the sentence referring to ROC curves.

Answer: Removed sentence referring to ROC curves.

5. In the results section, lines 140-144 are strangely placed. Consider moving them before the paragraph currently starting line 132.

Answer: Lines 140-144 are notes summarizing table 2. Can move it as per reviewer’s comment.

6. In the discussion, I do not understand why the authors specify that they stopped their literature search in June 2015. This does not make sense in 2020. Moreover, some articles evaluating the DBS techniques of interest have been published since. These articles should also be regarded especially since some articles fou,d similar results with the Roche technique (e.g. Taieb et al. OFID 2016).

Answer: Have included more recent studies and references as suggested. Pooled estimates of sensitivity and specificity have been changed to include more recent estimates reported in the World Health Organisation HIV molecular diagnostics toolkit to improve access to viral load testing and infant diagnosis 2019.

7. I still have issues with the Bland-Altman analysis. The figure should include the 95% confidence limits of the differences. I doubt the two horizontal grey lines correspond to these limits. The figures must be amended.

Answer: Have amended figures as advised.

(1) https://www.who.int/diagnostics_laboratory/evaluations/pq-list/hiv-vrl/170830_amended_final_pr_0145_027_00.pdf

---

## [Decision Letter · Decision Letter 2]

14 Apr 2020

Comparison of Alere q whole blood viral load with DBS and plasma viral load in the classification of HIV virological failure

PONE-D-19-29100R2

Dear Dr. Khan,

We are pleased to inform you that your manuscript has been judged scientifically suitable for publication and will be formally accepted for publication once it complies with all outstanding technical requirements.

With kind regards,

Julie AE Nelson, PhD

Academic Editor

PLOS ONE

Additional Editor Comments (optional):

Reviewers' comments:

Reviewer's Responses to Questions

**Comments to the Author**

1. If the authors have adequately addressed your comments raised in a previous round of review and you feel that this manuscript is now acceptable for publication, you may indicate that here to bypass the “Comments to the Author” section, enter your conflict of interest statement in the “Confidential to Editor” section, and submit your "Accept" recommendation.

Reviewer #1: All comments have been addressed

2. Is the manuscript technically sound, and do the data support the conclusions?

Reviewer #1: Yes

3. Has the statistical analysis been performed appropriately and rigorously? 

Reviewer #1: Yes

4. Have the authors made all data underlying the findings in their manuscript fully available?

Reviewer #1: Yes

5. Is the manuscript presented in an intelligible fashion and written in standard English?

Reviewer #1: Yes

6. Review Comments to the Author

Reviewer #1: (No Response)

7. PLOS authors have the option to publish the peer review history of their article (what does this mean?). If published, this will include your full peer review and any attached files.

Reviewer #1: No

---

## [Editor Report · Acceptance letter]

15 May 2020

PONE-D-19-29100R2 

Comparison of Alere q whole blood viral load with DBS and plasma viral load in the classification of HIV virological failure 

Dear Dr. Khan:

I am pleased to inform you that your manuscript has been deemed suitable for publication in PLOS ONE. Congratulations! Your manuscript is now with our production department. 

With kind regards,

on behalf of

Dr. Julie AE Nelson 

Academic Editor

PLOS ONE